# Mechanisms of Rapid Karyotype Evolution in Mammals

**DOI:** 10.3390/genes15010062

**Published:** 2023-12-31

**Authors:** Emry O. Brannan, Gabrielle A. Hartley, Rachel J. O’Neill

**Affiliations:** 1Department of Molecular and Cell Biology, University of Connecticut, Storrs, CT 06269, USA; emry.brannan@uconn.edu (E.O.B.); gabrielle.hartley@uconn.edu (G.A.H.); 2Institute for Systems Genomics, University of Connecticut, Storrs, CT 06269, USA

**Keywords:** chromosome evolution, karyotype evolution, chromosome rearrangements, speciation, cytogenetics, evolutionary breakpoints, centromere repositioning, satellite DNA, transposable elements, tachytely

## Abstract

Chromosome reshuffling events are often a foundational mechanism by which speciation can occur, giving rise to highly derivative karyotypes even amongst closely related species. Yet, the features that distinguish lineages prone to such rapid chromosome evolution from those that maintain stable karyotypes across evolutionary time are still to be defined. In this review, we summarize lineages prone to rapid karyotypic evolution in the context of Simpson’s rates of evolution—tachytelic, horotelic, and bradytelic—and outline the mechanisms proposed to contribute to chromosome rearrangements, their fixation, and their potential impact on speciation events. Furthermore, we discuss relevant genomic features that underpin chromosome variation, including patterns of fusions/fissions, centromere positioning, and epigenetic marks such as DNA methylation. Finally, in the era of telomere-to-telomere genomics, we discuss the value of gapless genome resources to the future of research focused on the plasticity of highly rearranged karyotypes.

## 1. Introduction

The organization of DNA into chromosomal domains is a fundamental genomic feature vital for critical regulatory processes. Such processes include the separation of active euchromatin from repressed heterochromatin, the formation of topologically associated domains (TADs) implicated in gene regulation, and the modulation of chromatin compaction and accessibility [1]. Despite the importance of genome organization to its function, rates of chromosomal rearrangements have been observed to vary significantly in a lineage-specific manner; some lineages have experienced rapid chromosomal changes that delineate species, while others have maintained a conserved karyotype across long evolutionary timeframes [2,3,4,5]. However, the basis for rapid chromosome evolution, which can contrast high levels of chromosome conservation, has yet to be determined. Recent improvements in sequencing technologies have corroborated and expanded upon findings made during the cytogenetic boom; chromosome-scale analyses with single nucleotide resolution promise to provide new insights into the genomic landscape of karyotypically divergent genera.

Taxonomic families such as Equidae [2], Hylobatidae [3], Macropodidae [4], Potoroidae [5], Ctenomyidae [6], Phyllostomidae [7,8], and Cervidae [9] have experienced rapid chromosome evolution with rates of rearrangement in these lineages proceeding at rates more frequent than other mammals. Several mechanisms have been proposed to explain this rapid evolution, including centromere repositioning [10], satellite expansions [11], and the mobility of transposable elements (TEs) [12]. Here, we explore the concept of tachytely, a term coined by paleontologist George Gaylord Simpson in 1944 to explain a rate of evolution faster than the modal rate for a lineage, in the context of chromosome rearrangements [13,14]. We apply these principles to the chromosome and nucleotide level through a modern lens, as improvements to sequencing technology and genome assembly provide the enhanced ability to perform genome-wide comparative analyses of inter- and intra-chromosome organization. Finally, we discuss the implications of next-generation and long-read sequencing on the continued examination of karyotype evolution.

## 2. Tachytely: A Retrospective Application to Rapid Karyotype Evolution

In 1944, paleontologist George Gaylord Simpson sought to understand differential rates of evolution among species using variations in the fossil records from different taxa [13,14]. In his seminal work “Tempo and Mode in Evolution”, Simpson recognized three classes of evolution, finding the rates of evolution to be highly variable even between closely related species, yet dependent upon “ecological opportunities” [13,14]. One such class defined by Simpson was *tachytely*, or exceptionally rapid change leading to a new adaptive zone, corresponding to evolutionary bursts or surges of short duration, and occurring among geographically isolated populations [13,14]. In contrast, *horotely* refers to the average rate of evolution, a spectrum that reflects a unimodal histogram and is characteristic of the evolution of many taxa, especially predatory mammals. Lastly, *bradytely* is characterized by arrested or prolonged evolution, typical of species identified as living fossils, such as horseshoe crabs [15] and opossums [16]. The extremely low rates of evolution represent a qualitatively distinct category from the rapid evolutionary rates of tachytely, yet are not an outlier of the moderate distribution of horotelic evolutionary rates [13,14]. Often characterized by morphological stasis, recent studies have examined the molecular stability of proposed bradytelic organisms, finding that low intraspecific diversity is not always correlative to a low rate of mutation [17,18].

These three classifications require a distinct combination of genetic factors regarding how they may yield speciation, phyletic, and quantum evolution; where speciation is considered a low-level process of generating diversity, with no significant input to trends or other larger-scale patterns; where phyletic evolution is a form of directional change, leading to evolutionary trends; and where quantum evolution is rapid and rare, analogous to Wright’s model of genetic drift [19]. Simpson posited that rates of evolution can differ from group to group, even among closely related lineages [13]. While these initial observations and conclusions predate the emergence of the modern field of comparative genomics, the concepts can be retrospectively applied to karyotype evolution, as the delineation of differential rates of chromosomal evolution could not be examined with fossil records nor limited molecular cytogenetics. In this review, we examine these concepts in view of species evolution concomitant with chromosome rearrangements, focusing on lineages that have undergone rapid changes to chromosome conformation analogous to Simpson’s initial view of tachytely.

## 3. Old and New Techniques in Studying Chromosome Evolution

With the expansion of genomic methodologies in the latter half of the 1900s, the emergence of various molecular and cytogenetic techniques finally enabled an improved genomic perspective of evolutionary rates and corresponding karyotypic changes. Conventional cytogenetic techniques have been used to scan for large-scale gains and losses and rearrangements among chromosomes. For example, chromosome banding can provide a simultaneous snapshot of the genome-wide copy number and structural variation (Figure 1A); however, it is also constrained due to ambiguous banding patterns and low resolution, limited by the complex way DNA is packaged into chromosomes, and limited by the requirement for intact chromosomes with distinct morphologies. Other staining techniques, such as silver staining of nucleolus organizer regions (NORs) [20,21] and C-banding to detect heterochromatin [22], offer the ability to track large blocks of repetitive DNA across chromosomes, but cannot delineate sequence changes among these regions [20,21,22]. One of the first significant cytogenetic techniques to emerge was the development of in situ hybridization using 3H-labeled nucleic acids; however, the radioactive isotope limited the widespread use of this technique [23]. To allow for a more ubiquitously available technique, fluorescence in situ hybridization (FISH) was developed [24], in which DNA was labeled with a fluorophore to emit fluorescence when hybridized to specific DNA sequences. FISH can be used with probes for specific genetic loci, such as telomeres or sites of known gain or loss of sequence.

Chromosome painting [25,26,27], a FISH method wherein DNA probes are derived from individual flow-sorted or microdissected chromosomes, offers a nuanced characterization of chromosome rearrangements (CRs) within metaphase spreads and interphase nuclei. When coupled with specific probes derived from BACs, chromosome painting synergistically enhances the precision of CR analysis, as illustrated in Figure 1B. This combined approach facilitates the tracking of rearrangements based on chromosomal origin and contributes to the generation of high-quality gene maps. However, despite their undeniable utility, these techniques face inherent limitations in their capacity to comprehensively interrogate an entire genome and detect small, intrachromosomal rearrangements. The intricacies of repetitive regions and the challenges posed by complex karyotypes with suboptimal morphology often present obstacles to the effectiveness of these cytogenetic methods. There is a growing need for innovative strategies to address these challenges and further elevate the precision and comprehensiveness of chromosomal assessment techniques to fully understand the complex mechanisms responsible for chromosome change and speciation. 

Though limited in scalability and cost, the development of Sanger sequencing [28] (first-generation sequencing) spearheaded a massive migration to next-generation sequencing (NGS) that offered parallelization, sequencing millions of contiguous segments simultaneously. In addition to short-read technologies (first- and next-generation sequencing), long-read (third generation) sequencing revolutionized the field of genome biology as improvements in read length allowed for more precise mapping of repetitive regions to be obtained for comparative analyses (Figure 1C,D). The commercialization of third-generation sequencing over the past decade has allowed for real-time sequencing of DNA fragments tens of thousands of nucleotides in length for the first time, alleviating many of the challenges associated with short-read genome assembly [29]. Such read lengths offer an improvement to assembling highly repetitive regions like the centromere [30]—a vital locus for cell survival and maintenance of genetic material—and chromosomal breakpoints [31], which can often be dense with repetitive elements. 

In addition to the generation of de novo genome assemblies, long-read based genomics combined with cytogenetic approaches enables the evolutionary history of chromosomal rearrangements to be traced within a group of organisms (Figure 1D). Furthermore, it enables the inference of evolutionary processes and patterns, such as evolutionary breakpoint regions (EBRs) [32], epigenetic marks [33], adaptive trait loci [34], and the evolution of specific traits involved in taxonomic or phenotypic diversification [35]. Although a remarkable amount of data has accumulated on the rates of cytogenetic change among many different lineages (e.g., [36,37]), the underlying molecular mechanisms that contribute to such change have still yet to be fully elucidated (but see [38,39] for examples). Combined with traditional cytogenetic approaches, emerging sequencing methods have improved our ability to understand the mechanisms underlying karyotype evolution, particularly those that defy average rates of evolution.

## 4. Mechanisms of Rapid Karyotype Evolution

Chromosome numbers constitute a taxonomic characteristic representative of species among a particular lineage. In mammals, it has been estimated that rates of chromosomal evolution average ~2 breakpoints per million years [40], with the lowest observed rates in eutherians occurring at a frequency of less than 0.2 breakpoints per million years [40]. Thus, most mammalian lineages are horotelic, with some showing bradytelic karyotypic stability over long evolutionary time periods, such as the Felidae and delphinids [41]. However, rapid karyotype evolution—i.e., tachytelic—at a rate exceeding average can play a critical role in speciation by creating a reproductive barrier among groups with divergent karyotypes. Several mechanisms defining karyotypic diversity have been described, including centromere repositioning, expansions of satellite DNA, mobile DNA, and other chromosome rearrangements (Figure 2).

### 4.1. Centromere Repositioning

The position of every centromere, the chromosomal site of kinetochore assembly, is epigenetically determined [42,43,44,45] and stable in a fixed location, promoting genome integrity. Centromere repositioning occurs when a neocentromere, or new centromere, emerges at an ectopic chromosomal location in tandem with the inactivation or loss of the native centromere (Figure 2I) [46]; this is reviewed in [10]. Such events are hypothesized to provide a rescue effect for native centromere inactivation and can create evolutionary new centromeres (ENCs) that become fixed in a particular lineage [47], rapidly evolving to orchestrate the recruitment of kinetochore proteins. While native centromeric DNA sequences are typically highly repetitive and are defined predominantly by satellite DNAs organized into higher order arrayed structures [48,49,50,51], centromere sequences diverge dramatically among lineages [52], neocentromeres, and ENCs [53,54,55]. As such, it is important to note that it is not the DNA sequence that denotes the centromere; instead, the incorporation of centromeric protein A (CENP-A), a variant of histone H3, into chromatin determines the active centromeric site [43,44,56,57]. While the underlying DNA sequence at ENCs can diverge from their native centromere counterparts, it has been observed in humans that centromere repositioning can also alter the underlying chromatin architecture, leading to the recruitment of RNA polymerase II and the initiation of active transcription concomitant with open chromatin in new centromere locations [58].

Centromere repositioning and subsequent neocentromere formation is implicated in creating karyotypic diversity, with heterozygotic meiosis caused by centromere repositioning, providing a reproductive barrier among closely related individuals that may contribute to sympatric speciation [59]. Centromere repositioning events are especially prevalent in species complexes characterized by rapid karyotype evolution, including some primates such as lemurs [60], squirrel monkeys [10,38], marmosets [38,61], macaques [62,63,64], the lesser apes, and gibbons [10,63]; several Macropodid species [65]; species across the Artiodactyla [41]; and the family Equidae [66]. For example, nearly half of the autosomal centromeres in macaque are defined as ENCs [64]. Similarly, in a study of *Equus* species, eight centromere repositioning events were identified since the divergence of zebra (*Equus burchelli*) and donkey (*Equus asinus*) from horse (*Equus caballus*), an evolutionary timespan of only two million years [10]. These findings provide evidence that this type of chromosomal rearrangement may be a feature of rapid chromosome evolution in some mammalian lineages.

### 4.2. Satellite DNA (satDNA)

Satellite DNAs are abundant tandem repeats that are classified into several categories based on repeat length: microsatellites (<10 nucleotide monomers), minisatellites or variable number tandem repeats (VNTRs) (~10–100 nucleotide monomers), and macrosatellites (the largest type of satellite, with >100 nucleotide monomers extending to several kilobases) [11,67]. Typically, copy numbers of satellites are highly polymorphic among individuals, contributing to variation within a population and species-specific diversity [68] (Figure 2II). As such, related orders may share an ancestral set of conserved satDNA families, yet have differential amplification among species, with high copy numbers in certain lineages contrasting low-copy counterparts found in other related species. For example, the involvement of satDNA in speciation by reproductive isolation has been implicated in the insect model organism *Drosophila melanogaster*, wherein differential amplification of a satDNA repeat, Zhr, on the X-chromosome has been shown to cause embryonic lethality of female hybrids between *D. melanogaster* and *D. simulans* [69]. Such examples of satDNA facilitating reproductive isolation have also been observed in several other species, including nematodes [70], beetles [71], spiders [72], ants [73], and certain plants [74]. 

Because of the highly complex, repetitive nature of satellites, improvements to long-read sequencing technologies continue to increase our ability to elucidate the structure and sequence evolution of conserved satellite sequence motifs. For example, [75] the satDNA, WalbRep, is only observed in the red-necked wallaby, (*Notamacropus rufogriseus*), originating from the walb endogenous retrovirus [75]; it is thought that walbRep was inserted into a pigment-related gene of a body color mutant wallaby [75,76]. WalbRep consists of the LTR (0.4 kb) and internal region (2.4 kb) of a nonautonomous walb copy [76]. The formation of walbRep occurred in the red-necked wallaby after it last shared a common ancestor with its sister taxon, the tammar wallaby (*Notamacropus eugenii*) in a time span of only 3 Mya [75].

### 4.3. Transposable Elements (TEs) 

Transposable elements (TEs), or mobile DNA, were first described by Barbara McClintock in 1950 as she studied the Ds transposons in maize [77]. First denoted as “jumping genes” for their ability to move from one locus in a genome to another, TEs are still commonly considered selfish elements, requiring a critical balance between a TE’s propensity for movement and the maintenance of genome integrity. Transposable elements can be classified in two ways based upon their methods of transposition: retrotransposons, or class I elements, require the reverse transcription of RNA into DNA to transpose via a copy-and-paste mechanism, while DNA transposons, or class II elements, move via a cut-and-paste mechanism and do not require an RNA intermediate (see [78] for a review). TEs can also be classified as autonomous or nonautonomous, wherein nonautonomous TEs lack the genetic elements capable of self-transposition and require borrowed elements, while autonomous TEs can self-extract and move around the genome. 

The movement of TEs are an important source of variation within a genome, wherein TEs can alter gene expression [79,80], disrupt coding genes [81], or promote recombination [82], allowing for dramatic and rapid restructuring of the genome that may exceed the changes offered by point mutations [83] (Figure 2III). These changes may allow populations to explore a fitness landscape more fully in a shorter period of time, increasing the adaptability of the population [84,85,86]. An example of this phenomenon is demonstrated by the TE dynamics in *Cardiocondyla obscurior* (tramp ant) [87]. Schrader et al. compared the genomes of two invasive species of *C. obscurior* to identify signatures of divergence on a genomic level and to determine how the species can rapidly adapt to different habitats; their findings elucidated phenotypic differences between the populations and a strong correlation between accumulations of TEs (‘TE islands’) and genetic variation [87]. However, TE mobility can also cause genomic instability when their expansion is unfettered or when they cause insertional mutations in genic and regulatory regions [88]. Thus, TE insertions can cause changes to regulatory networks and gene expression, providing selective advantages or instability. Moreover, TEs can facilitate non-homologous recombination events and generate reproductive barriers through chromosomal rearrangements. For example, in the phyllostomid bat *Tonotia saurophila*, a substantial centromeric enrichment of a *Tonotia*-specific LINE-1 partial ORF2 sequence has been observed, yet is not observed in other phyllostomine bats [7]. This observation is concomitant with a karyotype change from 2*n* = 32 observed in most Phyllostominae species to 2*n* = 16, suggesting that the LINE-1 expansion may play a role in the rapid karyotype evolution identified in this lineage [8].

### 4.4. Other Chromosome Rearrangements (CRs)

CRs are changes to any DNA segment that alters the native chromosome, such as fusions, fissions, insertions, deletions, duplications, or translocations (Figure 2IV). Chromosome fusion and fission events are primary mechanisms of karyotype evolution, as changes in chromosome numbers can create karyotypic heterozygosity, a reproductive barrier leading to reduced recombination and sterility of heterozygotes [89,90,91]. The altered recombination rate that can be a result of chromosome fusions can have permanent downstream consequences, affecting both the efficacy of purifying selection [92] and the impact of selection on linked sites, which consequently determines levels of genetic diversity [93,94]. For example, in the tuco-tuco rodent genus *Ctenomys*, diploid numbers range from 2*n* = 10 to 2*n* = 70, with extensive centric fusions and fissions, translocations, and differential repetitive content (i.e., C-banded material) [95]. While fertility among heterozygote carriers was unaffected, low effective population sizes coupled with altered recombination at CR breakpoints, often at centromeres, may have downstream consequences such as impacting selection of linked sites and levels of gene flow, further accelerating speciation (reviewed in [6]). Additionally, evolutionary fragile sites in the genome are often reused in chromosomal breakage and subsequent rearrangements of large chromosomal segments, also known as evolutionary breakpoints, or hotspots [31,96,97,98]. These evolutionary breakpoint regions (EBRs) are known to be non-randomly distributed in mammals, as it has been found that there are a limited number of regions in mammalian genomes that can be disrupted without negative consequences [98]. Over evolutionary time, these sites can become “recycled” in different species [99,100], being reused as breakpoints in the derivation of new karyotypic forms. Detection and mapping of these breakpoint regions are vital for both functional and comparative genomic purposes, especially when tracking evolution across divergent mammalian taxa. 

## 5. Lineages Characterized by Rapid Chromosome Evolution

### 5.1. Equidae

The genus *Equus* includes a diverse family composed of horses (*Equus caballus* and *Equus przewalskii*), African asses (*Equus asinus*), Asiatic asses (*Equus hemionus* and *Equus kiang*), and zebras (*Equus grevyi*, *Equus burchelli*, and *Equus zebra*) [101] (Figure 3A,B). All species are herbivorous and are primarily grazers; the domestic horse and donkey exist worldwide, whereas wild equine populations are limited to Africa and Asia. *Equus* karyotypes underwent rapid evolution after divergence from the common ancestor approximately 4.0–4.5 Mya [102,103]. The most recent radiation events, found among asses and zebras, occurred less than 1 Mya, and many species and subspecies emerged in this very short evolutionary time [103,104]. 

Rapid evolution of *Equus* genomes has been studied using bacterial artificial chromosome (BAC) probes, sub-chromosomal region-specific paints using FISH [66,105,106], phylogenetic mapping [107], and next-generation sequencing methods [2,108]. The numerous speciation events that occurred during the evolution of *Equus* species were accompanied by extensive karyotype reshuffling due to both chromosome rearrangements and centromere repositioning [66,103,105,106,109]. These karyotypic changes were responsible for a reduction in chromosome number and for the shift from an ancestral karyotype, with the majority of acrocentric chromosomes fusing to form submeta- and meta-centric chromosomes. As such, *Equus* diploid numbers range from 2*n* = 66 in the Przewalski’s horse (*Equus przewalskii*) and 2*n* = 64 in the domestic horse (*Equus caballus*) to 2*n* = 32 in the Hartmann’s mountain zebra (*Equus zebra*) [2,105].

During the evolution of extant equids, a total of 53 fusion events are estimated to have occurred [106], concomitant with at least eight centromere repositioning events [66]. Occurring at an exceptional rate, at least five centromere repositioning events occurred in the donkey post-divergence from the zebra, a timespan of only 1 Mya [66]. The location of such ENCs is variable among extant equids [66], with several having no satellite DNA near the neocentromere. For example, ChIP (chromatin immunoprecipitation)-on-chip experiments revealed that the ENC of horse chromosome 11 (ECA11) is completely devoid of satellite DNA, segmental duplications, and protein coding genes [110]. Despite being functional and stable in all horses observed, this neocentromere was not observed to be flanked by repeat rich satDNA [110]. Centromeric domains were then analyzed in the donkey by ChIP-seq, demonstrating that more than half of the centromeres (16 out of 31) are also devoid of satDNA [105,111]. The 16 satellite-free donkey centromeric domains were derived from centromere repositioning events that occurred in this lineage since they are orthologous to horse non-centromeric sequences [111]. Confirmation of these findings indicates that various centromeres within the *Equus* genus lack satellites entirely, establishing a distinctive model for mammalian centromeres [105,110,111,112,113,114,115]. It was hypothesized that there was not enough evolutionary time for maturation of these new centromeres and subsequent accumulation of satellite DNA [111]; however, chromosome 11 (ECA11) is satellite-free and fixed within the species [110], providing a unique opportunity to elucidate elements that drive ENCs using newer sequencing technologies.

Beyond the extraordinary number of ENCs, the high plasticity of equid genomes is also evidenced by the architectural organization of pericentromeric and centromeric satellite DNA families [116]. Three main satellite DNA families that were discovered include: 37cen, 2PI, and 137sat [105,116,117]. These satellite families differ in the length of their repeat unit, with 37cen consisting of a 221 bp repeat [105,116,117], 2PI of a 23 bp repeat, and 137sat of a 137 bp repeat [118]. While the centromeres of most equid genomes are satellite-rich, the centromeres of zebras and asses are mostly satellite-free with many satellite arrays at non-centromeric loci. Horses, on the other hand, display a unique satellite-free centromere coexisting with the typical satellite-rich mammalian centromeres. High-resolution FISH on combed DNA fibers demonstrated that at least some horse satellite-based centromeres may carry a mosaic arrangement of the different satellite DNA families where short arrays of the 2PI and EC137 satellites are closely intermingled and immersed within very large stretches of the 37cen sequence [118]. This organization suggests that recombination events among centromeric and pericentromeric satellite DNA can occur in the horse genome. The arrays of 37cen embed the centromeric core of horse satellite-based centromeres. Indeed, immunoprecipitation experiments with an anti-CENP-A antibody showed that this satellite family is the only one bound by CENP-A and thus bears the centromeric function [116]. Although centromeric satellites are typically AT-rich [30], 37cen is GC-rich, indicating that GC richness is compatible with the centromeric function—another contradiction to the established model for mammalian centromeres.

Lastly, some studies have found that repetitive sequences are associated with breakpoints and genomic fragility. However, equids did not show significant differences in repetitive sequences among species [2]. It has been hypothesized that mobile element insertions may play a more critical role in shaping the genome, following the discovery that L1 and ERV1 may have contributed to these rearrangements, as their presence is increased in rearrangement regions [2]. The discovery of an increased presence of L1 and ERV1 elements within rearrangement regions lends strong support to McClintock’s theory, suggesting that these elements have played a role in shaping the genome.

### 5.2. Hylobatidae

Lesser apes, or gibbons, are a group of 18–20 Hylobatidae species that last shared a common ancestor with great apes (humans, chimpanzees, bonobos, gorillas, and orangutans) roughly 17 Mya (Figure 3A,B) [119]. Native to southeast Asian tropical forests across Bangladesh, India, China, and Indonesia, gibbons are composed of four genera that underwent rapid differentiation from a shared ancestor roughly five million years ago. One of the most distinct features of the gibbon lineage is their propensity for interchromosomal rearrangements and variable diploid chromosome numbers among genera: *Hoolock* has a diploid number of 2*n* = 38; *Nomascus* has a diploid number of 2*n* = 52; *Symphalangus* has a diploid number of 2*n* = 50; and *Hylobates* has a diploid number of 2*n* = 44 [119]. The highly variable karyotypes of extant gibbons are derived from over 40 interchromosomal rearrangements, a rate of rearrangement 20 times higher than other primates, including: peri- and paracentric inversions, fusions, fissions, and Robertsonian translocations [120]. In stark contrast to the single large-scale interchromosomal rearrangement in great apes (a Robertsonian fusion of two ancestral chromosomes at the telomere to form human chromosome 2) [121,122], a combination of chromosome painting and BAC-FISH revealed an estimated 33 rearrangements in the ancestral Hylobatidae species. In a lineage-specific manner, such rearrangements have also led to 22 *Hoolock*-specific rearrangements and the formation of six ENCs (four specific to *Hoolock*), as well as 14 *Symphalangus*-specific rearrangements and the formation of three ENCs (one specific to *Symphalangus*) [120]. In addition, in *Hylobates*, chromosome rearrangements were identified to be enriched in heterochromatin [123]. As such, gibbons present a unique opportunity to study rapid chromosome evolution and centromere variation in the context of primates [3].

Despite the high rate of chromosomal rearrangements identified in gibbon species, no one causative factor has been identified; however, the propagation of LAVA, an active gibbon-specific composite retrotransposon composed of LINE, *Alu*Sz, VNTR, and *Alu*-like segments [124,125,126] is proposed to play a role [126]. LAVA elements, although non-autonomous, are still active within gibbon genomes and utilize L1 machinery to mobilize [127]. While present in all gibbon species, LAVA has propagated at variable rates among genera and are most commonly positioned in the centromeres of *Hoolock* species [3,125]. In fact, while most primates have centromeres rich in 171-bp alpha satellite repeats [48,128,129,130], gibbon centromeres are highly variable and transposable element-rich. For example, while *Nomascus* and *Symphalangus* appear to have alpha satellite-rich centromeres [131,132], *Hoolock* and *Hylobates* centromeres are not predominantly enriched with alpha satellites, and large arrays of LAVA elements surround most *Hoolock* centromeres [3,125]. 

In addition to serving as a source of variation, LAVA elements in gibbons are also a key contributor to the hypothesis of repeat-mediated karyotype evolution by exaptation [126]. While most transposable element insertions provide neutral sources of variation or are selected against due to disruption of functional DNA, the process of exaptation involves the co-option of insertions to form favorable outcomes, such as performing a regulatory function or providing a transcription factor binding site to affect gene regulation [12,85]. Most human-specific regulatory elements have proven to be young, lineage-specific transposable element insertions [133], not unlike the insertion patterns displayed by LAVA elements. In the *Nomascus* genus, more than half of identified LAVA insertions are in or near genes, particularly those related to the cell cycle and chromosome segregation [119]. Additionally, in a study of nine *Hoolock* individuals, LAVA insertions were enriched near genes with GO terms relating to DNA repair and centrosome regulation, and genes near such insertions were shown to be more highly expressed using RNA-sequencing [119,126]. Furthermore, it was determined that six transcription factor binding sites are enriched within LAVA, including PU.1, STAT3, SRF, SOX10, SOX17, and ZNF143 [126]. With more than half of LAVA insertions being lineage-specific (compared to only 16.5% of insertions shared among genera), it is reasonable to suspect that such variable insertions may contribute to karyotype variation via polymorphic control of cell cycle genes [125,126].

In a study of chromosomal breakpoints across gibbon individuals, it was determined that breakpoints are enriched with segmental duplications, simple repeats, and *Alu* elements [124]. *Alu* elements within 150 base pairs of breakpoints were determined to have two times higher CpG content yet lower CpG methylation compared to other *Alu* elements, making them morphologically distinct from *Alu* elements found elsewhere in the genome [124]. Since transposable elements are typically methylated to suppress transcription and their subsequent proliferation within a genome [134], it is hypothesized that these undermethylated *Alu* elements provide a higher degree of open chromatin compared to methylated *Alu*s, leading to lineage-specific rearrangements via non-homologous end joining, non-allelic homologous recombination, or an unknown mechanism [124,135]. Paired with observations in other lineages with highly derived karyotypes, such as the identification of a hypomethylated retroviral element, kangaroo endogenous retrovirus (KERV)—with higher replication in wallabies concomitant with centromere expansion [136,137]—such features are plausible drivers of karyotype variation. 

### 5.3. Macropodidae

Marsupials (Metatherians) are one of two groups of Therian mammals from which the Prototherians, or monotremes, last shared a common ancestor approximately 180 Mya [138]. Eutherians and the Metatherians diverged nearly 160 Mya (Figure 3A,B) [138]; Metatheria includes marsupials that possess a pouch and give birth to partially developed, or altricial, offspring. There are around 300 extant living species of marsupials, which are grouped into three American orders (*Didelphimorphia*, *Microbiotheria*, and *Paucituberculata*) and four Australasian orders (*Dasyuromorphia*, *Diprotodontia*, *Notoryctemorphia*, and *Peramelemorphia*) [139]. The family *Macropodidae* belongs to the Australasian order *Diprotodontia*, including kangaroos, wallabies, possums, koalas, wombats, and many others. 

Until recently, the vast majority of studies of marsupial chromosome evolution have focused purely on a cytogenetic perspective, which granted an overview of large-scale rearrangements. In addition, FISH mapping of genes has been used to resolve breakpoints and identify more minor inversions. However, these approaches are limited in their ability to interrogate whole genomes [5,37,65,98,140,141,142]. The macropodid family has undergone considerable genomic reshuffling amongst its more than 60 species. Chromosome complements observed amongst marsupials are easily derived from the predicted marsupial ancestral karyotype through various combinations of fissions, fusions, translocations, and centromere repositioning [5]. The conserved segments were identified by using chromosome paints from three Australian species of *Diprotodontia*: the tammar wallaby (*Macropus eugenii*, 2*n* = 16), the brushtail possum (*Trichosurus vulpecula*, 2*n* = 22), and the long-nosed potoroo (*Potorous tridactylus*, 2*n* = 12, XX F, 2*n* = 13, XY1Y2M) [5]. These cross-species paints detected 15 conserved chromosome segments and their arrangement in the 2*n* = 14 ancestral karyotype was subsequently rebuilt from G-banding studies [5]. Additionally, chromosome painting of distantly related species has shown that marsupial chromosomes are divided into 19 conserved segments, with 18 autosomal segments and one corresponding to the X chromosome [142,143]. The macropodiformes’ ancestral karyotype can be derived from the 2*n* = 14 ancestral marsupial complements by considerable reshuffling, a combination of five fissions, one fusion, inversions, and centric shifts [5]. Surprisingly, even species within the same genus possessing the same diploid number and displaying morphologically similar chromosomes do not have the same arrangement of these 18 conserved syntenic blocks, undergoing independent Robertsonian fusions of different ancestral chromosomes. 

The ancestral marsupial is proposed to have possessed a 2*n* = 14 karyotype, similar to the karyotype observed in many species distributed across the marsupial phylogeny (Figure 4A) [5]. For example, Dasyurids, a lineage which includes over 70 extant species divided into 17 genera, all maintain the ancestral chromosome state (Figure 4B). In contrast, closely related Macropodids have undergone genomic reshuffling since they diverged from a common ancestor approximately 23 Mya [144] (Figure 4C). The large chromosomes of marsupials are predicted to have emerged from fusion events in the therian ancestor. Conversely, a more complex set of rearrangements, including a series of fission and fusion events, are predicted to have led to a higher chromosome number in the ancestral eutherians [144]. Diploid numbers of macropodids range from 2*n* = 10 F, 11 M for the swamp wallaby (*Wallabia bicolor*) to 2*n* = 24, for the banded hare wallaby (*Lagostrophus fasciatus*) [144]. 

The role of centromeric DNA in the rapid evolution of macropodid genomes has yet to be thoroughly studied. Although there was success in reconstructing phylogenetic relationships among species based on chromosome evolution [4,139,145], the observation that many species within the *Macropus*/*Notamacropus*/*Wallabia*/*Osphranter* group have experienced breakpoint reuse between syntenic blocks [98]—all of these at active centromere locations—in the derivation of a novel karyotype presented challenges. Moreover, data from interspecific marsupial hybrids indicate that the centromere contributes to chromosome diversity and can undergo dramatic shifts of genome architecture within one generation [146]. The retroviral sequence KERV was identified in a hybrid at centromeric loci that had undergone demethylation and subsequent KERV amplification, resulting in chromosome remodeling [136]. By utilizing cross-species chromosome painting on a macropodid hybrid, each chromosomal rearrangement was found to be entirely restricted to the centromere [136]. Thus, the centromeres of these hybrids appear to be hotspots of genome instability [98]. The centromeric rearrangements could result from the transposition of TEs or the amplification of centromeric sequences in cis. Such a dramatic increase in de novo chromosome rearrangements in these hybrids suggests that a reorganization of the karyotype can occur rapidly [136].

Interestingly, there is a strong correlation among centromere satDNA, breakpoint reuse, and karyotype convergence amongst nine macropod species, with contractions and expansions of predominant satellites occurring with specific chromosome rearrangements [98]. Further examination of some of the hybrids revealed destabilization of the centromeres, resulting in chromosome rearrangements such as fissions, whole-arm reciprocal translocations, and the formation of minichromosomes [141], as well as the amplification of repetitive sequences associated with macropodid centromeres [136,141]. A similar hotspot preference for ENCs has been found in non-hybrid species when considering synteny across the phylogeny. Comparative sequence analysis in the tammar wallaby (*N. eugenii*) of a native centromere site, an evolutionary breakpoint associated with previous centromere activity and the potential for new centromere formation [65,98], revealed an enrichment for LINEs and endogenous retroviruses at this breakpoint [100]. EBs tracked within the class Mammalia harbor sequence features retained since the divergence of marsupials and eutherians that may have predisposed these genomic regions to large-scale chromosomal instability [100]. The frequent generation of de novo chromosome rearrangements in the Macropod family, along with the breakpoint re-usage at centromeric sites riddled with repetitive elements, provides an obvious route for dramatic changes in karyotype within one generation.

### 5.4. Potoroidae

Potoroidae are a small family of diprotodont marsupials closely related to the Macropodidae family. The Potoroidae family is endemic to Australia and Tasmania, and includes nine species placed in five genera, composed of potoroos, rat-kangaroos, and bettongs. The differences between familial diploid numbers are very distinct in this family, as the lowest number is seen in *Potorous tridactylus*, the long-nosed potoroo, with a diploid number of 2*n* = 12, XX F, 2*n* = 13, XY1Y2 M, and the highest number in *Aepyprymnus rufescens*, the Rufous bettong, with a diploid number of 2*n* = 32. 

Although the XX/XY sex chromosome system is the most common among Therian species, it is not the only sex chromosome system found. Ohno’s law posited that translocations between the X and autosomes would be selected against to avoid disrupting the dosage compensation mechanism [147]. Notably, translocations or fusions between autosomes and the sex chromosomes have been observed in several mammalian species. For example, the large X-chromosome seen in some marsupials, such as *P. tridactylus* or *W. bicolor*, is formed by the fusion of an autosomal segment to the conserved X segment [143]. 

Multiple sex chromosome systems were discovered in marsupials, and these male-specific sex chromosomes include: a metacentric X-chromosome, an acrocentric Y-chromosome, and a second very small Y-chromosome. During meiosis in the male, the three sex chromosomes form a trivalent, called the dense plate, which orients so that the two separate Y-chromosomes would orient to one pole and the X-chromosome to the other [148]. Translocations and/or fusions between the sex chromosomes and autosomes (SA fusions) are widespread amongst marsupials [149]. An example is the swamp wallaby (*W. bicolor*, 2*n* = 10, XX F, 2*n* = 11, XY1Y2M), where chromosome paints derived from the tammar wallaby revealed that the short arm of the swamp wallaby X chromosome was homologous to the tammar wallaby X, whereas the long arm shared homology with tammar wallaby chromosomes 2 and 7, as does Y2, representing the autosome to which the X was fused. 

Linking sexually antagonistic alleles to sex chromosomes can increase the average fitness of both sexes; therefore, SA fusions are predicted to be more common than fusions joining two autosomes (AA-fusions), contradicting Ohno’s predictions [92]. One group studied the probability of SA fusions for any XY sex chromosome system with any number of autosomes [150]. They found that when an organism’s autosome number is small, a large proportion of fusions are expected to be SA-fusions, even under a null model that assumes they are not selectively favored [150], thus promoting speciation through the generation of a multiple-sex chromosome system. In fact, for an XY sex chromosome system, the probability of a given fusion being an SA-fusion does not drop below 25% until the diploid autosome count is greater than or equal to 16 [150]. With macropods and potoroos having low diploid numbers, the observation that many species have multiple sex chromosome systems fits predictions of this model. 

Among the most distantly related eutherians, gene content and order of the X chromosomes are typically highly conserved, with rodents being the exception, as rodents appear to have undergone rearrangements in gene order [151,152,153]. In contrast, gene order is not conserved between marsupials and eutherians nor between different marsupial species [154], with a high degree of rearrangement observed between opossum, Tasmanian devil, and tammar wallaby X chromosomes [154,155]. Although marsupials also inactivate one X chromosome in female somatic cells, the mechanism differs from that observed in eutherian mammals [156], and may be more tolerant of intrachromosomal rearrangements.

### 5.5. Artiodactyla: Cervidae

Order *Artiodactyla*, or even-toed ungulates, is a large mammalian order that includes whales, pigs, hippos, camels, and other ruminants. Studies utilizing routine and differential staining techniques have highlighted remarkable karyotype uniformity among cetaceans, with a consistent diploid chromosome number of 2*n* = 44 across most species [157,158,159,160,161]. Chromosome maps for Odontoceti (toothed) species, such as the Atlantic bottlenose dolphin, pilot whale, and Yangtze finless porpoise, revealed identical karyotypes among these species, emphasizing the stability and low rates of karyotype evolution in cetaceans [162]. 

In contrast, within Artiodactyla, the family Cervidae is remarkable among mammals for the extent of the genome-wide chromosomal diversification among species (e.g., [9,163,164]), with diploid numbers ranging from 2*n* = 6/7 to 2*n* = 70. Cervidae is the second most diverse in the suborder *Ruminantia*, which is made up of bovines, sheep, giraffes, deer, and others. The family consists of two subfamilies: *Cervinae,* which includes Old World deer, comprised of muntjac, elk, red deer, and fallow deer; and *Capreolinae*, or New World deer, comprised of roe deer, reindeer, and moose. Members of Cervidae are widespread in America and Eurasia and have high economic and ecosystem values. 

Though most members of the Cervidae family have a very high diploid number, with most species observed with (2*n*) ranging from 32 to 70 chromosomes, there is a wide modality found within the family, with one of the most interesting examples of rapid karyotype evolution found in muntjacs*. Muntiacus muntjak*, the Indian muntjac, has the lowest diploid number recorded for all mammals, with 2*n* = 6, XX F and 2*n* = 7, XY1Y2 M, once again demonstrating that mammals with low diploid numbers are more susceptible to SA fusions [150]. In contrast, the highest diploid number in the Cervidae family is found in *Muntiacus reevesi*, the Reeve’s or Chinese muntjac, with a 2*n* = 46 karyotype. It has been proposed that the ancestral karyotype is 2*n* = 70, which is like that of *Hydropotes inermis* (water deer) [165], and recurrent chromosome fusions have led to the karyotypes of extant muntjac species varying from 2*n* = 46 of *M. reevesi* [166], to 2*n* = 6/7 of *M. muntjak* [167]. 

Using chromosome painting [164,168] and FISH [169,170], it was determined that muntjac karyotypic diversity arose primarily through centromere-telomere tandem fusions and, to a lesser extent, centromere–centromere fusions [163]. One group traced the changes in the muntjac karyotypes using chromosome painting between *M. reevesi* and other ruminants, as well as bacterial artificial chromosomes (BACs) mapped by FISH between *M. muntjak* and *M. reevesi* [171]. As further support for the tandem fusion theory, several sequence-based studies have found evidence for the juxtaposition of centromeric repeats and telomeric sequences at fusion sites [9,163]. These fusion events probably account for the drastic reduction of chromosome numbers seen in the Indian muntjac. It was shown later by chromosome painting and BAC clone mapping in black muntjacs [172] that the X chromosome was translocated to the autosomal chromosome 4, and that of the other chromosome 4 was translocated to the short arm of chromosome 1 followed by an inversion, which resulted in the generation of a multiple sex chromosome system [2]. High-resolution, BAC-FISH X chromosome maps across Artiodactyla revealed inversions and centromere repositioning were key rearrangements during the dynamic evolution of the X chromosome [96]. In fact, the rate of X-specific rearrangements in this order significantly surpasses that among eutherian mammals, with nine paracentric inversions, two pericentric inversions, and five centromere reposition events identified in X chromosome evolution across 18 species [173]. Comparatively, the eutherian and artiodactyl ancestral X chromosomes differed only by one small inversion, with an additional rearrangement proposed to derive the Ruminantia ancestral X (RAX). 

Studies have additionally produced a chromosome-level assembly for both *M. muntjak* and *M. reevesi*, enhancing the investigation of muntjac karyotype evolution through comparative genomics [2]. Although many insights into muntjac genome evolution have been obtained through cytogenetic analyses, the two chromosome-scale genome assemblies enabled a comparative analysis of intra-chromosome organization and gene evolution in muntjacs and confirmed the evolutionary sequence of fissions and fusions described cytogenetically [2]. It was found that chromosome segments in cervids and cow have remained highly collinear since their divergence ~20 million years ago [2]. This, in turn, implies that the translocations and fusions observed in the muntjacs were not accompanied by major inversions or other internal rearrangements, though neither the repetitive terminal regions of chromosomes nor the fusion junctions were studied [2]. Additionally, the same group determined that a tenfold acceleration in the rate of chromosome change occurred in the *M. muntjak* lineage relative to the mammalian average [2]. Other muntjac species that more recently diverged from the *M. muntjak* branch also have unique rearrangements [172,174], suggesting that the fusions in this lineage did not occur all at once but rather as separate events. To search for genetic changes correlated with rapid karyotype evolution, genes with accelerated rates of evolution in *M. muntjak* were identified as potential candidates involved in chromosome maintenance, although many genes with signals of rapid evolution had no obvious relationship to chromosome biology [2].

Genome assemblies also provide the ability to examine topologically associated domains (TADs) (Figure 1C), or megabase-scale genomic regions that interact within a compartment but have few interactions outside of this chromatin domain [175]. Often, such breakpoint regions are more commonly found at TAD boundaries than within TADs, suggesting an evolutionary constraint for the maintenance of these interacting compartments [176,177]. However, the chromosome fusion events in muntjacs have little impact on the compartment type and TADs, even near fusion sites, but rather can lead to novel significant interactions connecting distant genomic loci or across the fusion sites. Such interactions may have both cellular and morphological significance during the evolution and adaptation of muntjac deer.

## 6. Conclusions

The patterns of chromosome number and arrangement of closely related species are a defining feature amongst organisms, yet the mechanisms through which some species retain an ancestral karyotypic state while others undergo rapid karyotypic radiation are not well understood. Several factors described herein have been proposed to contribute to such rapid chromosomal reshuffling, including centromere repositioning, satellite expansions, and transposable element-mediated chromosome rearrangements. However, without high quality, error-free genome assemblies of these rearranged genomes, the contributing factors leading to tachytelic karyotype rearrangement have proved difficult to elucidate; repetitive breakpoints defining chromosome rearrangements in lineages with rapid karyotype evolution have often been excluded from genome assemblies, requiring these analyses to take a genome-independent approach. Recent improvements to long-read sequencing technologies and assembly methodologies [178,179] are poised to propel the field of comparative genomics into a new era, providing resolution of previously intractable genomic regions implicated in the rapid karyotype evolution observed in particular lineages: repeats such as satellite DNA and TEs, centromeres, and EBRs [180].

Historically, the evolutionary perspective of Simpson’s tachytely evokes a gradualistic approach to the examination of evolutionary rates. Built largely upon the examination of fossil records, the postulation of tachytely as unusual and rapid evolutionary development of a species resulting in an adaptive zone shift was initially considered through the lens of morphological characteristics. While posited prior to the emergence of modern comparative genomics and sequencing technologies, the concepts can be retrospectively applied to karyotype evolution, providing a distinguishable perspective on species with tachytelic evolution. For example, Simpson recognized the rapid evolution of humans in comparison to other apes and prosimians; yet, from a karyotypic perspective, *Homo sapiens* and other great apes have undergone fewer large-scale chromosome rearrangements throughout evolutionary time than their more recently diverged lesser ape counterparts. Such interpretive variation highlights the essential demand for emerging comparative genomic approaches, such as gapless genome assemblies, in the examination of karyotype and chromosome evolution of species.

As long-read sequencing technology becomes more ubiquitous, high quality, telomere-to-telomere, gapless assemblies have become a more accessible and feasible approach for both model and non-model species alike. In response, the number of collaborative consortiums with goals to produce complete, error-free reference genomes has steadily increased. The Telomere-to-Telomere (T2T) Consortium, once a group with the singular goal of producing an error-free haploid human reference, has expanded to include a wide variety of human samples and species, including primates, drosophila, ruminants, and others [181]. Due to these efforts, seven T2T reference genomes are already available for study, with dozens more in curation [182]. The Earth BioGenome Project (EBP), described as a “moonshot for biology”, aims to sequence all eukaryotic biodiversity within the next 10 years [183]. Others, including the Vertebrate Genomes Project (VGP) [184], Deep Ocean Genomes (DeepOGen) [185], the Darwin Tree of Life (DToL) [186], and Oz Mammal Genomics (OMG) [187], represent just a fraction of the targeted groups working to sequence organisms in different ecological niches. In a new era of emerging high quality reference assemblies, the capabilities of comprehensive comparative genomics to investigate karyotypic evolution has become increasingly feasible. As the prevalence and accessibility of telomere-to-telomere genome assemblies continues to increase for model and non-model species, so too will our understanding of the vast network of genomic and epigenetic factors contributing to centromere specification and chromosome evolution.

## Figures and Tables

**Figure 1 genes-15-00062-f001:**
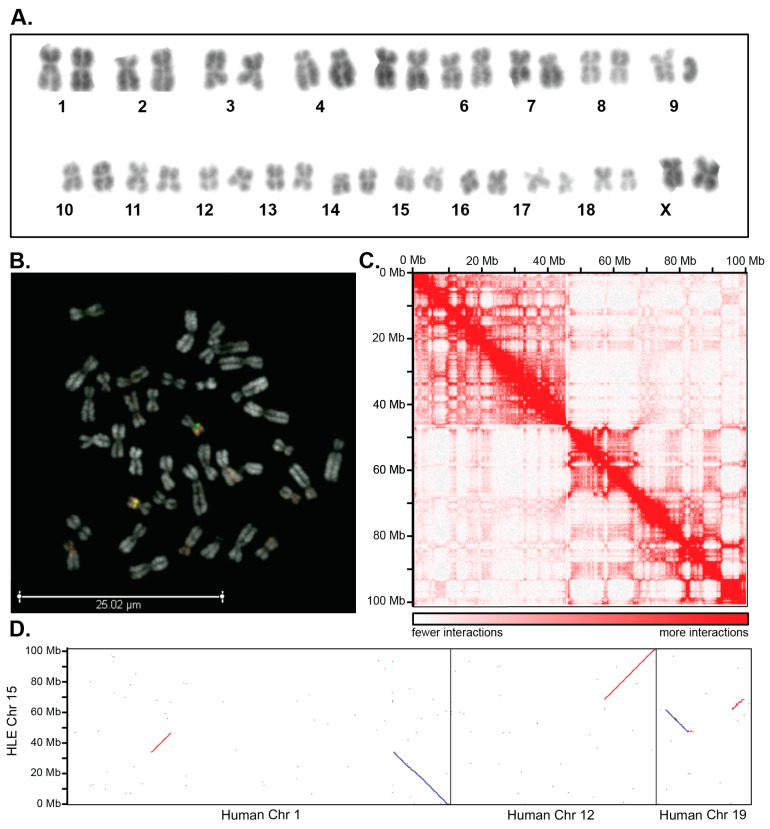
Sequencing combined with cytogenetic techniques improves the resolution of chromosome rearrangements. (**A**) A karyotype for a female Eastern hoolock gibbon, *Hoolock leuconedys* (2*n* = 38). (**B**) A human locus probe for chr19p13.2 (green) and chr19q13.33 (red) on *Hoolock leuconedys* metaphase chromosome spreads reveals hybridization of both to the q arm of chromosome 15, suggesting a rearrangement in the *Hoolock* chromosome compared to human chromosomes. (**C**) An Omni-C contact map for chromosome 15 of the genome assembly *Hoolock* shows long-range chromatin interactions that form a characteristic plaid pattern of two mutually excluded (**A**,**B**) compartments. A-compartments correspond to gene-rich and active chromatin, while B-compartments are primarily enriched in repressive chromatin. Genomic coordinates are indicated on both axes, and a color code represents the contact frequency between regions on a scale where dark red represents high contact frequency and white represents low contact frequency. Combining assembly and contact frequency provides a more robust visualization of breakpoints involved in the chromosome rearrangement. (**D**) An alignment dot plot between assembled human (CHM13) chromosomes (horizontal) and *Hoolock* chromosome 15 (vertical) allows for accurate identification of the precise breakpoints along the *Hoolock* chromosome and depicts regions of synteny among the chromosomes (red line represents complementary alignments, blue line represents alignments in the reverse complement).

**Figure 2 genes-15-00062-f002:**
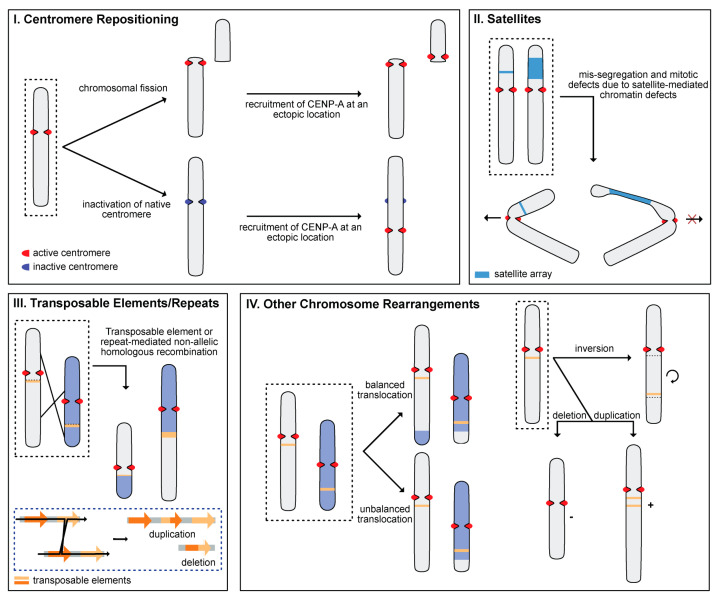
Mechanisms of karyotype evolution. (**I**) Centromere repositioning can occur when a native centromere actively recruiting CENP-A (red) becomes inactivated (dark blue) through DNA- (loss of centromeric sequence) or epigenetic- (loss of CENP-A incorporation) mediated events, or due to a chromosomal fission event wherein one chromosome segment loses the native centromere. In both cases, centromere repositioning occurs when CENP-A is recruited and fixed to an ectopic location which now serves as the functioning centromere. (**II**) satellite expansions (blue) can mediate karyotypic evolution via mis-segregation and mitotic defects caused by defects to chromatin structure. Large satellite expansions can produce chromatin defects, causing chromatids to fail to segregate (bottom right). (**III**) Transposable elements and other repeats (orange and brown) sharing sequence identity across chromosomes can result in non-allelic homologous recombination (NAHR), wherein unequal crossing over results in chromosome modifications across regions that are not alleles (white chromosome and blue chromosome are not haplotypic). Below, a zoomed panel (blue box) shows a possible NAHR event occurring between two transposable elements (orange and brown arrows), leading to duplication and deletion events on different chromosomes. (**IV**) Other chromosomal mechanisms of rearrangement can contribute to karyotypic variation, including chromosome inversions, deletions, and duplications. The fate of segments from two non-allelic chromosomes (white and blue) are indicated. Red dots demarcate the CENP-A-delimited centromere and orange lines represent a locus.

**Figure 3 genes-15-00062-f003:**
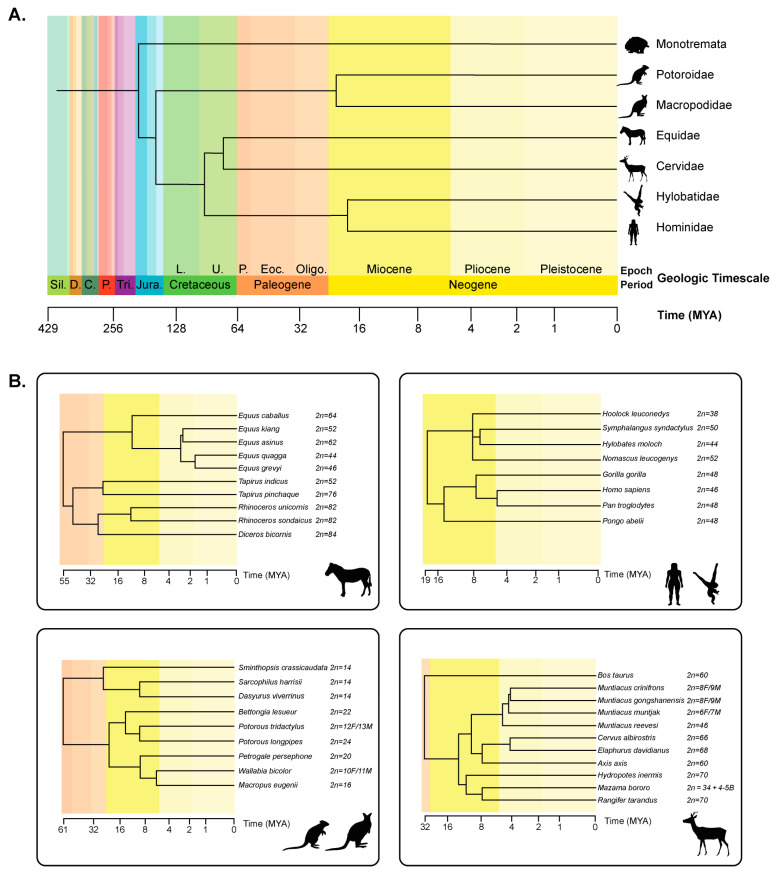
Phylogeny of lineages displaying rapid karyotype evolution. (**A**) A phylogeny of lineages with rapid karyotype evolution are depicted, contrasted by the Hominidae family (great apes), which have maintained a karyotype with few large-scale chromosome rearrangements since their divergence from lesser apes ~17 million years ago (Mya). Below, the geologic timescale of divergence is shown, with epochs and periods depicted by different colors. A timescale is depicted in millions of years. In the evolutionary timescale, abbreviations refer to the following: in the Paleogene Period, abbreviations refer to the Paleocene, Eocene, and Oligocene epochs; in the Cretaceous Period, abbreviations refer to the Lower and Upper epochs; and abbreviations in the remaining periods refer to, from most to least recent, the Jurassic, Triassic, Permian, Carboniferous, Devonian, and Silurian periods. (**B**) A detailed phylogeny of selected species within lineages described herein is depicted, including Equidae (top, left), Hylobatidae and Hominidae (top, right), Macropodidae and Potoroidae (bottom, left), and Cervidae (bottom, right). As above, various colors depict geologic epochs and a timescale is displayed below each phylogeny. Diploid chromosome numbers for each species are included, highlighting the wide range of chromosome number variation across the evolution of these lineages. Phylogenies in (**A**,**B**) were created using TimeTree [104].

**Figure 4 genes-15-00062-f004:**
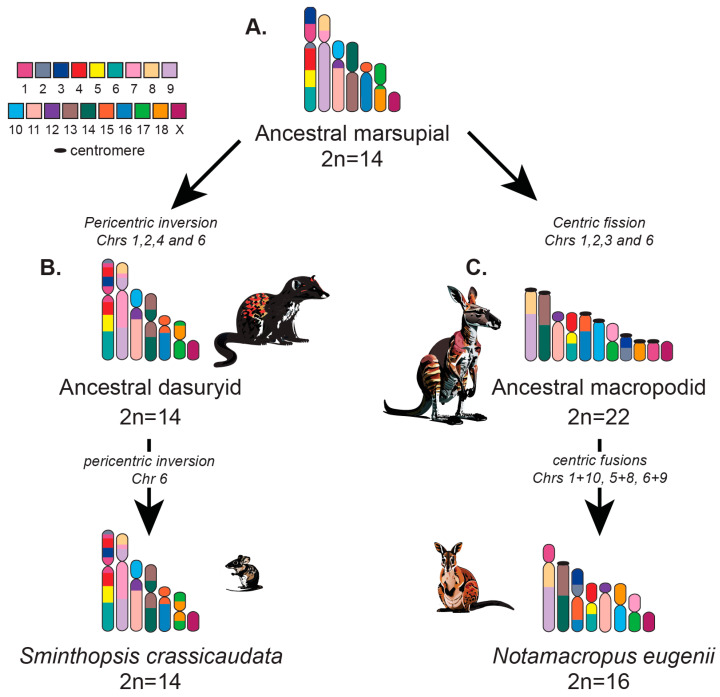
Ancestral karyotypes and conserved syntenic blocks for marsupial chromosomes. (**A**) The ancestral marsupial karyotype is composed of 18 conserved autosomal blocks (left, colored key) and one sex chromosome block found in all marsupials as suggested by [135]. (**B**) Ancestral dasyurid karyotype, 2*n* = 14, and derived *Sminthopsis crassicaudata* karyotype, 2*n* = 14. (**C**) Ancestral macropodiform, 2*n* = 22, and derived *N. eugenii* karyotypes, 2*n* = 16. Karyotype construction adapted from [5], with rearrangements leading to each lineage indicated.

## Data Availability

Not applicable.

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
