# Peer review of "Mechanisms of Rapid Karyotype Evolution in Mammals"

_genes, 2023, doi:10.3390/genes15010062_

Round 1

Reviewer 1 Report

Comments and Suggestions for Authors

Brannan and colleagues provide a comprehensive review of concepts and mechanisms of karyotypic evolution in mammals. While the title and text address crucial conceptual issues and outline directions and perspectives within the current understanding for the group as a whole, I miss the inclusion of other mammalian groups in the discussion.

How the authors determined which groups underwent rapid karyotypic evolution and consequently were included in the discussion, in detriment to other groups. In fact, some mammalian groups, such as felids in general, tend to exhibit a relative karyotypic stability. However, groups like bats and even whales show remarkable karyotypic diversity.

In the case of whales, althought the 2n varies only between 42 and 44, numerous karyotypic formulas and morphologies of sex chromosomes has been described. Processes, such as fusions, translocations, and inversions, are attributed to this karyotypic diversity in these groups. It would be interesting to at least mention these other mammalian groups throughout the discussion.

Overall, the manuscript is well-written and deserves publication. However, a few minor adjustments need to be addressed.

In Figure 1, the panel A is empty.

In Figure 2, there are characters that looks like a configuration error. Also, I did not understand exactly what the black rectangles in the panels of the image signify, especially the size difference between them in different panels. Do the black rectangles represent the ancestral condition for each scenario presented? The authors could clarify this in the image caption or directly on the image.

Throughout the text, there are some typos also some empty spaces between words.

The resolution of the Figure 3 must be improved.

Comments on the Quality of English Language

The english is fine. Only minor typos.

Reviewer 2 Report

Comments and Suggestions for Authors

In this review, the authors summarized 10 lineages prone to rapid karyotypic evolution. I found the manuscript very interesting and well-Written! Congratulations! The study falls into the scope of Genes and thus could be published. 

However, I am curious as to why rodents were not encompassed in this review. The chromosome organization within this taxonomic group exhibits an even greater degree of diversity than some of the groups highlighted in this manuscript. For instance, the genus Ctenomys showcases an unparalleled spectrum of chromosomal variation among mammals, ranging from 2n = 10 to 2n = 70. Other example is the range of 2n on Neacomys genus (28 to 64). Such pronounced diversity (and other  cases) merits consideration within the context of this review and could significantly enrich the discussion on karyotypic evolution. 

Reviewer 3 Report

Comments and Suggestions for Authors

Authors provide a nice review on mechanisms of rapid karyotype evolution in mammals

Comments:

-          Beside the main topic authors also make comments on technical aspects on how to study genomes and refer already in abstract to the value of gapless genome resources. As these are (concerning heterochromatic regions) not to be expected to be really available soon, this needs to be avoided and/ or clearly stated if authors mean here the brand new approaches which also can access heterochromatin, or talk about gapless euchromatic genome resource

-          Section 3: please do not mention PRINS and fluorescent polymerase chain reaction (PCR=  – these were a techniques applied in the 1990s and no one uses it actually any more – or make clear that they are outdated – only FISH is left and used and extremely helpful in karyotype characterization. It is just a matter of having a specialized lab, which can do the tests (PMID: 35360869). It is not more sophisticated than NGS-approaches are. Specifically, FISH in combination with microdissection is ground breaking in chromosomal evolution research still (e.g. PMID: 37233068)

-          Line 98-107: G-banding is not possible in most species – instead CBG and NOR staining are much more relevant in evolution research – please include and discuss

-          Figure 1A – there is just a grey field visible there – please check

-          Lines 125 ff: as stated – if one would microdissect the corresp. chromosomes the wcp probes would be available - and this has been done – see PMID: 19635709 – alike, if there is a BAC bank of the species available it is no problem to use those probes in FISH (PMID: 11272792) – besides there is always the possibility of cross-species FISH using probes from human or other species available (PMID: 8793207) – overall there are <50 papers where FISH was used in Equidae in Pubmed; so please make this more prominent – mol cytogenetics is still extremely helpful and necessary to align sequencing with cytogenetic data.

-          For Hylobatidae and chr-studies see also PMID: 26893612, PMID: 26893612, PMID: 26893612

-          Overall, please provide a clear take home message at the end of the paper and the abstract – what is the result / are the results of this review in terms of “Mechanisms of Rapid Karyotype Evolution in Mammals”? This is not clear yet.

Round 2

Reviewer 3 Report

Comments and Suggestions for Authors

Paper is more clear and better understandable now

still Fig. 1A is again not visible